# Risk factors and clinical features for pulmonary paragonimiasis-associated pneumothorax

**Yunhong Song**[1☉], **Jeongmin Lee**[1☉], **Wonchang Hahn**[1☉], **Yujeong Jang**[1☉], **Seungwon Na**[1☉], **Sang-Min Oh**[2,3,4], **Joo-Hee Hwang**[2,3,4], **Chang-Seop Lee**[2,3,4], **Yeong Hun Choe**[2,3,4]*, **Jeong-Hwan Hwang** [2,3,4]*

**1** Jeonbuk National University Medical School, Jeonju, Jeonbuk, Republic of Korea, **2** Department of Internal Medicine, Jeonbuk National University Medical School, Jeonju, Jeonbuk, Republic of Korea, **3** Research Institute of Clinical Medicine of Jeonbuk National University, Jeonju, Jeonbuk, Republic of Korea, **4** Biomedical Research Institute of Jeonbuk National University Hospital, Jeonju, Jeonbuk, Republic of Korea

☉ These authors contributed equally to this work.
* cd4tcell@hanmail.net(YHC); smilehwang77@hanmail.net(JHH)

## Abstract

### Background

Pulmonary paragonimiasis, a food-borne zoonotic helminthiasis, is a parasitic disease of the lung caused by infection with trematodes species of the genus *Paragonimus*. Although pneumothorax has been reported as occuring with paragonimiasis, to date no study has been performed concerning the clinical features and predictive risk factors for this condition.

### Methods

This retrospective study, which aims to fill this gap, was conducted at Jeonbuk National University Hospital. All patients (aged ≥19 years) were diagnosed with paragonimiasis between May 2011 and December 2021. Medical records were reviewed and information concerning age, sex, vital signs, underlying diseases, clinical signs and symptoms, laboratory findings, radiologic findings, treatment, and clinical outcomes was collected. An odds ratio (OR) for the risk factors associated with pneumothorax was calculated using the binary logistic regression model.

### Results

Among 179 consecutive patients diagnosed with pulmonary paragonimiasis, the postive rate of pneumothorax was 10.6% (19/179). Pneumothorax occurred mostly in the right lung (78.9%, 15/19), and intrapulmonary parenchymal lesions showed an ipsilateral relationship with pneumothorax (94.7%, 18/19). Fifteen patients (78.9%, 15/19) of pneumothorax associated with pulmonary paragonimiasis are accompanied by pleural effusion. Most of patients with pneumothorax (89.5%, 17/19) underwent chest tube insertion as a first treatment. Three patients (15.8%) showed relapses but in no case was a death recorded. Asthma (odds ratio [OR] 8.10, 95% confidence interval [CI] 1.43–45.91), chest pain (OR 8.15, 95%

**Funding:** The author(s) received no specific funding for this work.

**Competing interests:** The authors have declared that no competing interests exist.

CI 2.70–24.58), and intrapulmonary lesions (OR 8.94, 95% CI 1.12–71.36) were independent risk factors for pulmonary paragonimiasis-associated pneumothorax.

## Conclusions

Our findings suggest that clinicians should keep in mind the possibility of pneumothorax when approached by patients with pulmonary paragonimiasis complaining of chest pain, accompanied by intrapulmonary lesions or with asthma as an underlying disease.

### Author summary

Paragonimiasis, as a typical food-borne zoonotic helminthiasis, is an important public health problem. Approximately one million people suffer from paragonimiasis annually and areas in which the disease is newly endemic are increasingly identified. Paragonimiasis causes complex symptoms in multiple organs, but chest symptoms are its most distinguishing clinical feature. Most symptomatic paragonimiasis infections appear as pulmonary diseases such as pneumonia, bronchitis, pleuritis with pleural effusion, and pneumothorax. Pneumothorax is a debilitating and expensive symptomatic pulmonary complication that can occur in cases of paragonimiasis. Therefore, it is necessary to be aware of the risk factors for pneumothorax in patients with pulmonary paragonimiasis and to evaluate the possibility of pneumothorax in those patients. Our study suggests that history of asthma, chest pain, and intrapulmonary lesions are clinically significant risk factors of pneumothorax associated with pulmonary paragonimiasis. Clinicians should approach patients with pulmonary paragonimiasis that complain of chest pain or have intrapulmonary lesions or asthma as an underlying disease with the possibility of pneumothorax in mind and should note that pulmonary paragonimiasis is one of the causes of the differential diagnosis of secondary spontaneous pneumothorax.

## Introduction

Pulmonary paragonimiasis, a food-borne zoonotic helminthiasis, is a parasitic disease of the lung caused by infection with trematodes species of the genus *Paragonimus*. Paragonimiasis is the second most common disease among food-borne trematodiasis, with over 50 species of *Paragonimus* having been identified throughout the world, mostly in Asia [1,2]. Only a few species of *Paragonimus* are responsible for most infections, with *P. westermani* among the most notable [1]. *P. westermani* is present primarily in Asia, and has been detected in China, the Philippines, Japan, Vietnam, South Korea, Taiwan, and Thailand [1,3]. Although a decreasing number of patients in Japan and Korea has led to less focus on paragonimiasis, its endemicity and re-emergence suggests that the disease deserves more attention [1].

Human infection begins with ingestion of metacercariae, the infective form for the mammalian host that live within crustaceans such as crab or crayfish [4]. These metacercariae excyst in the duodenum, and pass through the intestinal wall, peritoneal cavity, diaphragm, and pleural cavity to eventually enter the lung tissue, where they mature to adult flukes [4]. For this reason, most symptomatic paragonimiasis infections appear as pulmonary diseases such as pneumonia, bronchitis, pleuritis with pleural effusion, and pneumothorax [4,5]. The measured frequency of paragonimiasis-associated pneumothorax has varied from study to study,

with assessments producing results as divergent as 6.7% (2/30), 8.3% (3/36), 16.9% (75/443), 17% (12/71), and 25% (4/20) [4–8].

To date, no study has examined the clinical features and predictive risk factors of pneumothorax in patients with paragonimiasis-associated pneumothorax. The purpose of this study was to investigate the clinical features and these risk factors.

## Materials and methods

### Ethics statement

This study was conducted in accordance with the Declaration of Helsinki. This study was approved and conducted consistent with the guidelines established by the Institutional Review Board of the Jeonbuk National University Hospital (IRB no.: CUH 2023-07-021). Institutional Review Board of the Jeonbuk National University Hospital waived the need for specific informed consent from study participants because this study data was anonymized, so could not identify the subjects.

### Study design and data collection

This was a retrospective study conducted at Jeonbuk National University Hospital in Jeonju, Korea. Jeonbuk National University Hospital is a university-affiliated teaching hospital with approximately 1,200 beds and the largest referral center in Jeollabuk-do, a province of Korea. Patients aged 19 years or older diagnosed with paragonimiasis between May 2011 and December 2021 were included. Pulmonary paragonimiasis was diagnosed when one or more of the following diagnostic criteria were satisfied: seropositivity in ELISA (Genedia Ab ELISA, Green Cross MS, Yongin, Korea), an immunoserologic test for specific antibody for *P. westermani*, the detection of *Paragonimus* eggs (in sputum samples, bronchoalveolar lavage, pleural effusion, or feces) in cytology, and the identification of adult fluke or eggs in lung tissue obtained from lung biopsies [3,4]. We retrospectively reviewed the medical records of the patients and collected the following data: age, sex, vital signs, underlying diseases, clinical signs and symptoms, laboratory findings, radiologic findings, treatment, and clinical outcomes.

### Assessment of pneumothorax

The presence of pneumothorax was confirmed through chest X-ray. Chest CT scans were performed, unless patient rejected it, to identify underlying pulmonary diseases. Relapse of pneumothorax was defined as a case that occurred within two years from pulmonary paragonimiasis-related pneumothorax, taking into account the relevance of the current episode. A past history of pneumothorax was defined as an independent relationship with a length of more than two years from the time of occurrence of current pneumothorax associated with pulmonary paragonimiasis. Large-size pneumothorax was defined, consistent with British Thoracic Society (BTS) 2010 guidelines [9], as the presence of a visible rim of >2 cm between the lung margin and the chest wall (at the level of the hilum), and with American College of Chest Physicians (ACCP) guidelines as the presence of a distance greater than 3 cm from the apex of the collapsed lung to the ipsilateral thoracic cupola [10].

### Statistical analysis

Descriptive statistics include frequency analysis (percentages) for categorical variables and mean ± standard deviation or median (range) for continuous variables. We analyzed categorical variables using a chi-square test or Fisher's exact test; we analyzed continuous variables using Student's t-test or the Mann–Whitney U-test, as appropriate. We performed univariate

and multivariate logistic regression analyses to evaluate the risk factors for pneumothorax in patients with pulmonary paragonimiasis. Variables that showed a significant difference (P<0.1) in the univariate analysis were included in the multivariate analysis. P<0.05 was considered statistically significant, and all probability values were two-tailed. We used the variance inflation factor (VIF), and Cramer's V as statistical measures of detecting multi-collinearity among continuous variables or categorical variables, respectively. Point biserial correlation was used to evaluate multi-collinearity between dichotomous (nominal) and continuous variables. SPSS software version 27.0 (IBM Corp., Chicago, IL, USA) was used for all statistical analyses.

## Results

The baseline characteristics of the patients are presented in Table 1. During the study period, a total of 179 consecutive patients with pulmonary paragonimiasis were identified. All subjects showed positive results in the ELISA test, 15 patients had lung fluke or eggs in their lung tissue (as obtained from lung biopsy), and five had *Paragonimus* eggs confirmed through cytology performed on BAL, stool, or other samples. Of these patients, 19 (10.6%) were classified as having pneumothorax associated with paragonimiasis, with the remaining 160 (89.4%) not having pneumothorax. There was a significance difference in instances of asthma between the pneumothorax and non-pneumothorax groups (15.8% vs. 3.1%, p = 0.041). In addition, the frequency of previous instances of pneumothorax, chronic obstructive pulmonary disease (COPD), and lung operations was higher in the pneumothorax group than in the non-

**Table 1. Baseline characteristics of the patients included in the study.**

|  | Pneumothorax | Non-pneumothorax |  |
|---|---|---|---|
| Parameter | N = 19 | N = 160 | P value |
| Age (year) | 53±13 | 60±14 | 0.034 |
| Male | 14 (73.7) | 112 (70.0) | 0.739 |
| Height (cm) | 167.5±7.5 | 163.9±8.1 | 0.070 |
| Weight (kg) | 64.7±9.6 | 64.4±12.0 | 0.936 |
| BMI (kg/m$^2$) | 22.9±2.5 | 24.2±3.9 | 0.206 |
| <18.5 kg/m$^2$ | 1 (5.3) | 5 (3.1) | 0.495 |
| Smoking | 6 (31.6) | 57 (35.6) | 0.727 |
| Underlying disease |  |  |  |
| Previous pneumonia | 1 (5.3) | 21 (13.1) | 0.475 |
| Previous pneumothorax | 2 (10.5) | 2 (1.3) | 0.056 |
| Previous tuberculosis | 1 (5.3) | 22 (13.8) | 0.296 |
| Previous lung operation | 2 (10.5) | 6 (3.8) | 0.203 |
| COPD | 5 (26.3) | 33 (20.6) | 0.559 |
| Asthma | 3 (15.8) | 5 (3.1) | 0.041 |
| Lung malignancy | 0 (0.0) | 8 (5.0) | 1.000 |
| Concurrent tuberculosis lesion | 0 (0.0) | 2 (1.3) | 1.000 |
| Idiopathic pulmonary fibrosis | 0 (0.0) | 7 (4.4) | 1.000 |
| Chronic liver disease | 0 (0.0) | 15 (9.4) | 0.374 |
| Congestive heart failure | 0 (0.0) | 8 (5.0) | 1.000 |
| Chronic kidney disease | 2 (10.5) | 18 (11.3) | 1.000 |

Data are presented as mean ± SD or number (%).

Abbreviation: BMI, body mass index; COPD, chronic obstructive pulmonary disease

pneumothorax group, but here again the difference was not statistically significant. There was also no significant difference in smoking between the two groups.

Table 2 highlights the clinical features, radiologic findings, and laboratory findings associated with the pneumothorax and non- pneumothorax groups. There was a significant difference between the pneumothorax and non-pneumothorax groups in the number of subjects with chest pain and dyspnea (57.9% [11/19] vs 16.9% [27/160], p<0.001; 68.4% [13/19] vs 28.1% [45/160], p<0.001). The pneumothorax group showed a significantly higher prevalence of pleural effusion (78.9% vs 28.7%, p<0.001). Within this group, a right pleural effusion was the most common, appearing with a frequency of 46.7% (7/15). Intrapulmonary lesions were significantly higher in the pneumothorax group than in the non-pneumothorax group, but there was no difference between the two groups in detailed lesions. There was a significant difference in the eosinophil count and eosinophil percent between the two groups, (3.5±2.6 vs. 1.8±3.6, p = 0.045; 32.1±15.1 vs. 16.7±18.0, p<0.001).

Pneumothorax occurred mostly in the right lung (78.9%, 15/19), and intrapulmonary parenchymal lesions showed an ipsilateral relationship with pneumothorax (94.7%, 18/19) as shown in Tables 2 and 3. Applying the 2001 ACCP guidelines, a large-sized pneumothorax was present in 36.8% (7/19) of cases; applying the 2010 BTS guidelines, a large-sized

**Table 2. Clinical, laboratory, and radiological features of the patients enrolled in the study.**

|  | Pneumothorax | Non-pneumothorax |  |
|---|---|---|---|
| Parameter | N = 19 | N = 160 | P value |
| Clinical features |  |  |  |
| Symptomatic | 18 (94.7) | 122 (76.3) | 0.079 |
| Cough | 10 (52.6) | 63 (39.4) | 0.266 |
| Hemoptysis | 1 (5.3) | 31 (19.4) | 0.204 |
| Chest pain | 11 (57.9) | 27(16.9) | <0.001 |
| Dyspnea | 13 (68.4) | 45(28.1) | <0.001 |
| Fever | 2 (10.5) | 32 (20.0) | 0.535 |
| Gastrointestinal symptoms | 2 (10.5) | 33 (20.6) | 0.375 |
| Radiologic findings |  |  |  |
| Pleural effusion | 15 (78.9) | 46 (28.7) | <0.001 |
| Right | 7 (36.8) | 15 (9.4) | 0.003 |
| Left | 4 (21.1) | 11 (6.9) | 0.058 |
| Both | 4 (21.1) | 20 (12.5) | 0.292 |
| Intrapulmonary lesions | 18 (94.7) | 102 (63.7) | 0.007 |
| Consolidation | 11 (57.9) | 58 (36.3) | 0.067 |
| Cystic lesion | 3 (15.8) | 7 (4.4) | 0.076 |
| Nodules | 8 (42.1) | 52 (32.5) | 0.402 |
| Linear streak | 1 (5.3) | 4 (2.5) | 0.433 |
| Calcified lesion | 0 (0.0) | 11 (6.9) | 0.610 |
| Laboratory findings |  |  |  |
| WBC (1,000/mm$^3$) | 9.1±3.7 | 8.3±4.8 | 0.489 |
| Eosinophil (1,000/mm$^3$) | 3.5±2.6 | 1.8±3.6 | 0.045 |
| Eosinophil (%) | 32.1±15.1 | 16.7±18.0 | <0.001 |
| IgE[a] | 810.5±1038.9 | 1113.4±1055.6 | 0.312 |

Data are presented as mean±SD or number (%).

Abbreviation: WBC, white blood cell count; COPD, chronic obstructive pulmonoary disease; BMI, body mass index.

[a] The IgE test results were available in 16 of 19 patients in the pneumothorax group and 57 of 160 patients in the non-pneumothorax group.

**Table 3. Descriptive data on clinical features of 19 patients with pulmonary paragonimiasis-associated pneumothorax.**

| Case | Age | Dyspnea | Location | Size | | Treatment | | | Outcome | | |
|---|---|---|---|---|---|---|---|---|---|---|---|
| | Smoking | Chest pain | | ACCP | BTS | Oxygen | Chest tube | Tube duration | Recur[a] | Death | Hosptal duration |
| 1 | 52 years | Yes | Right | 4.4 cm | 1.6 cm | Yes | Yes | 7 days | No | No | 9 days |
| | No | Yes | | | | | | | | | |
| 2 | 66 years | Yes | Right | 2.3 cm | 0.5 cm | Yes | Yes | 3 days | No | No | 4 days |
| | Yes | No | | | | | | | | | |
| 3 | 59 years | Yes | Right | 2.5 cm | 1.8 cm | Yes | Yes | 5 days | No | No | 17 days |
| | No | Yes | | | | | | | | | |
| 4 | 54 years | No | Right | 1.9 cm | 0.4cm | Yes | Yes | 7 days | No | No | 10 days |
| | No | Yes | | | | | | | | | |
| 5 | 61 years | Yes | Right | 1.1 cm | 0.8cm | Yes | Yes | 4 days | Yes | No | 4 days |
| | No | No | | | | | | | | | |
| 6 | 75 years | Yes | Right | 1.7 cm | 2.9 cm | Yes | Yes | 3 days | No | No | 12 days |
| | Yes | No | | | | | | | | | |
| 7 | 75 years | No | Right | 1.0 cm | 0.2 cm | Yes | No | 0 | Yes | No | 0 |
| | Yes | No | | | | | | | | | |
| 8 | 52 years | Yes | Right | 2.5 cm | 0.8 cm | Yes | Yes | 7 days | No | No | 7 days |
| | No | No | | | | | | | | | |
| 9 | 50 years | No | Right | 2.7 cm | 1.2 cm | Yes | Yes | 3 days | No | No | 4 days |
| | No | Yes | | | | | | | | | |
| 10 | 49 years | Yes | Right | 4.9 cm | 1.5 cm | Yes | Yes | 3 days | No | No | 5 days |
| | No | Yes | | | | | | | | | |
| 11 | 39 years | Yes | Right | 0.5 cm | 0.2 cm | Yes | No | 0 | No | No | 0 |
| | Yes | Yes | | | | | | | | | |
| 12 | 51 years | Yes | Right | 4.0 cm | 0.6 cm | Yes | Yes | 6 days | No | No | 8 days |
| | No | No | | | | | | | | | |
| 13 | 43 years | Yes | Left | 8.9 cm | 2.1 cm | Yes | Yes | 3 days | No | No | 4 days |
| | No | Yes | | | | | | | | | |
| 14 | 35 years | No | Right | 2.9 cm | 1.1 cm | Yes | Yes | 3 days | No | No | 4 days |
| | Yes | Yes | | | | | | | | | |
| 15 | 49 years | No | Left | 4.9 cm | 1.2 cm | Yes | Yes | 3 days | No | No | 5 days |
| | No | Yes | | | | | | | | | |
| 16 | 56 years | Yes | Right | 1.1 cm | 1.3 cm | Yes | Yes | 3 days | Yes | No | 4 days |
| | No | Yes | | | | | | | | | |
| 17 | 31 years | Yes | Left | 3.3 cm | 0.2 cm | Yes | Yes | 3 days | No | No | 4 days |
| | No | Yes | | | | | | | | | |
| 18 | 33 years | No | Left | 4.3 cm | 2.1 cm | Yes | Yes | 16 days | No | No | 18 days |
| | Yes | No | | | | | | | | | |
| 19 | 72 years | Yes | Right | 0.8 cm | 1.1 cm | Yes | Yes | 3 days | No | No | 4 days |
| | No | No | | | | | | | | | |

Abbreviation: ACCP, the 2001 American College of Chest Physicians guideline; BTS, the 2010 British Thoracic Society guideline

[a] In the case of the three patients with relapsed pneumothorax, a chest tube was inserted in case 5, oxygen therapy was performed in case 7, and VATS was performed in case 16.

pneumothorax was present in 15.8% (3/19) of cases. Most of patients with pneumothorax (89.5%, 17/19) underwent chest tube insertion as a first treatment. Three patients (15.8%) showed relapses but in no case was a death recorded. The three patients who relapsed were

treated with VATS, chest tube, and oxygen supply, respectively, and no further relapses were observed.

After a univariate analysis, multi-collinearity was evaluated among independent variables. Age, height, dyspnea, pleural effusion, eosinophil (1,000/mm$^3$), and eosinophil (%) were removed. Previous pneumothorax (odds ratio [OR] 9.29, 95% confidence interval [CI] 1.23–70.26), asthma (OR 5.81, 95% CI 1.27–26.61), chest pain (OR 6.77, 95% CI 2.49–18.42), and intrapulmonary lesions (OR 10.24, 95% CI 1.33–78.66) were statistically significant (Table 4). A subsequent multivariate analysis suggested that asthma (OR 8.10, 95% CI 1.43–45.91), chest pain (OR 8.15, 95% CI 2.70–24.58), and intrapulmonary lesions (OR 8.94, 95% CI 1.12–71.36) were independent risk factors for pulmonary paragonimiasis-associated pneumothorax (Table 5).

## Discussion

Our study is the first to assess the clinical characteristics and risk factors of patients with pneumothorax associated with pulmonary paragonimiasis. Our study suggests that history of asthma, chest pain, and intrapulmonary lesions are clinically significant risk factors associated with pulmonary paragonimiasis.

Chest pain is observed in most cases of primary spontaneous pneumothorax (PSP), but is less common in cases of secondary spontaneous pneumothorax (SSP) [11]. Dyspnea, rather, is reported as the most common symptom in SSP [12,13]. In this study, dyspnea was the most common symptom, followed by chest pain and cough. However, chest pain was identified in 57.9% of patients with pneumothorax associated with paragonimiasis, and simultaneous chest pain and dyspnea were reported in 37%. According to Yoshida A, et al. chest pain occurs when the *Paragonimus* larvae invades and injures the pleura, causing pneumothorax [1]. Therefore, although chest pain is less common in SSP, it is nevertheless an important clinical clue that suggests pneumothorax in SSP related to pulmonary paragonimiasis.

In this study, asthma, rather than COPD, was identified as a significant risk factor for pneumothorax associated with pulmonary paragonimiasis. COPD is the lung disease most frequently identified as being associated with SSP [13]. Unlike COPD as an underlying disease related to SSP, according to previous reports, SSP has been reported as an uncommon complication of asthma [14]. As the invasion of the lung parenchyma and damage to the pleura by the *Paragonimus* larvae relates to the development of pneumothorax, we anticipated that

**Table 4. Univariate analysis to determine risk factors for pneumothorax associated with pulmonary paragonimiasis.**

| Variables | OR (95% CI) | P value |
|---|---|---|
| Age | 0.96 (0.93–0.99) | 0.037 |
| Height | 1.06 (0.99–1.12) | 0.073 |
| Previous pneumothorax | 9.29 (1.23–70.26) | 0.031 |
| Asthma | 5.81 (1.27–26.61) | 0.023 |
| Chest pain | 6.77 (2.49–18.42) | <0.001 |
| Dyspnea | 5.54 (1.98–15.46) | 0.001 |
| Pleural effusion | 9.29 (2.93–29.49) | <0.001 |
| Intrapulmonary lesions | 10.24 (1.33–78.66) | 0.025 |
| Eosinophil (1,000/mm$^3$) | 1.00 (1.00–1.00) | 0.064 |
| Eosinophil (%) | 1.04 (1.02–1.06) | 0.001 |

Abbrivation: OR, odds ratio; CI, confidence interval.

**Table 5. Multivariate analysis to determine risk factors for pneumothorax associated with pulmonary paragonimiasis.**

|  | Univariate analysis | | Multivariate analysis[a] | |
| --- | --- | --- | --- | --- |
| Variables | OR (95% CI) | P value | OR (95% CI) | P value |
| Previous pneumothorax | 9.29 (1.23–70.26) | 0.031 | | |
| Asthma | 5.81 (1.27–26.61) | 0.023 | 8.10 (1.43–45.91) | 0.018 |
| Chest pain | 6.77 (2.49–18.42) | <0.001 | 8.15 (2.70–24.58) | <0.001 |
| Intrapulmonary lesions | 10.24 (1.33–78.66) | 0.025 | 8.94 (1.12–71.36) | 0.039 |

Abbrivation: OR, odds ratio; CI, confidence interval.

[a] Age, height, dyspnea, pleural effusion, eosinophil (1,000/mm$^3$), and eosinophil (%) were excluded from the multivariate analysis due to multi-collinearity between independent variables.

COPD would show a stronger relationship to the occurrence of pneumothorax. In fact, asthma was identified as a significant risk factor for SSP associated with pulmonary paragonimiasis.

The mechanism of pneumothorax associated with pulmonary paragonimiasis has not been clearly elucidated. What is clear, however, is that larvae from the small intestine enter the abdominal cavity and penetrate the diaphragm, the pleura is damaged, and the larvae then migrates to the lung parenchyma, resulting in pneumothorax, which can be accompanied by pleural effusion due to an inflammatory response [15]. This pneumothorax was observed as an early manifestation of radiologic findings of pulmonary paragonimiasis [15]. In this study, 78.9% of patients with pneumothorax showed signs of pleural effusion, most of which were on the right side.

Pneumothorax appears to occur in association with a variety of parasitic diseases. Among parasitic infections of the lung, hydatidosis and ascariasis have been reported as causing pneumothorax [16]. Human hydatid disease is a parasitic infection caused by the larval form of *Echinococcus granulosus* characterized by cystic lesions in the lung, liver, and other parts of the body [17]. Hydatid cysts can perforate into the pleural cavity and cause pleural effusions, hydro-pneumothorax, simple and tension pneumothorax, and empyema [18]. Pneumothorax resulting from the rupture of the hydatid cyst into the pleural cavity is a rare complication [19]. In one study, the positive rate of pneumothorax in case of hydatidosis was assessed to be 6.2% [17]. In another study, the positive rate of pneumothorax in patients with hydatid lung disease was reported to be 3.3% (3/91) [20]. The positive rate of pneumothorax associated with pulmonary paragonimiasis has been reported to be 10.6%, which is higher than that of hydatidosis [17,20]. Tension pneumothorax, a serious but rare complication of pulmonary hydatid cysts, was not observed in our study. Pneumothorax has also been rarely reported in cases of ascariasis caused by *Ascaris lumbricoides*, the giant intestinal round worm [21,22]. In these cases, symptomatic pulmonary disease occurs after migration of *Ascarias lumbricoides* from the contaminated foods in the gastrointestinal tract to the pulmonary circulation [22]. It is characterized by eosinophilic pneumonia with allergic inflammation, in which pulmonary infiltrates in radiology are present along with respiratory symptoms such as fever, cough, dyspnea, wheezing, and airway obstruction [23]. In such cases, pneumothorax is likely to be caused by the mechanical action of the larva in association with an inflammatory response in the pleura, as well as the underlying pulmonary parenchymal tissue [22]. However, because most patients do not complain of respiratory symptoms, the presence of larvae maturation or migratory phases in the lung is a very uncommon finding [21,23]. Finally, schistosomiasis [24] and amobiasis [25] have been reported as uncommon causes of pneumothorax. The parasitic diseases mentioned above should be considered during the differential diagnosis of pneumothorax, especially in endemic areas.

Paragonimiasis, as a typical food-borne zoonotic helminthiasis, is recognized as a serious medical problem in some countries [1]. Approximately one million people suffer from paragonimiasis annually and areas in which the disease is newly endemic are increasingly identified [1]. In one meta-analysis paragonimiasis accounted for 26% of heavy and symptomatic infections among food-borne trematodiasis, making it the most common, and the DALY of paragonimiasis was reported to be second only to clonorchiasis among food-borne trematodiasis [2]. The non-negligible disability weight of DALY, wide regional distribution, and severe health consequences, and rise in the number of endemic regions, all suggest that paragonimiasis is an important public health problem that must not be neglected [26].

Pneumothorax is a debilitating and expensive symptomatic pulmonary complication that can occur in cases of paragonimiasis. According to one 12-year nationwide cohort study, the outpatient treatment period for spontaneous pneumothorax was 11 days, the hospitalization period was 14 days, with the outpatient treatment cost being $94.50, and the inpatient treatment cost was $2,523.00 [27]. With more than 20,000 new cases of spontaneous pneumothorax occurring each year in the United States, the health care system is burdened by nearly $130,000,000 in associated expenses [28]. The outcomes for secondary spontaneous pneumothorax are worse than those associated with primary spontaneous pneumothorax, with higher recurrence rates and longer hospital lengths of stay [29]. In the case of secondary spontaneous pneumothorax, the clinical course can be severe due to the underlying diseases, so hospitalization, close monitoring, and proper treatment is necessary [10]. In light of the individual burden of paragonimiasis, which is one of the causes of secondary spontaneous pneumothorax, and the associated system-wide high medical costs, national programs for paragonimiasis control should be implemented, similar to those in Korea and Japan [1,26]. In addition, for patients with paragonimiasis who have asthma, chest pain, or intrapulmonary lung lesions, which are the risk factors identified in this study, an evaluation for pneumothorax should be conducted.

This study had several limitations. First, it was a retrospective, single center study conducted in a small population. However, ours is the first to analyze pneumothorax associated with pulmonary paragonimiasis and to describe the associated risk factors and clinical features. The population size examined was larger than that of previous studies concerning pulmonary paragonimiasis [4,5,7,8]. In the future, additional research will be required to evaluate the causality of risk factors, and an even greater number of subjects should be used. Second, we were unable to evaluate all potential risk factors for pneumothorax related to pulmonary paragonimiasis that would have been accessible through an observational study. Third, we did not fully address selection bias. We could not confirm to include all patients having pulmonary paragonimiasis because we retrospectively analyzed data recorded by our hospital. Fourth, using this research design, it was not possible to distinguish between asthma and paragonimiasis as possible causes of the pneumothorax.

In sum, this study suggests that chest pain, asthma, and intrapulmonary lesion are clinically important predictive risk factors of pneumothorax associated with pulmonary paragonimiasis. Clinicians should approach patients with pulmonary paragonimiasis that complain of chest pain or have intrapulmonary lesions or asthma as an underlying disease with the possibility of pneumothorax in mind and should note that pulmonary paragonimiasis is one of the causes of the differential diagnosis of SSP.

## Supporting information

**S1 Data. Clinical raw data pulmonary paragonimiasis-associated pneumothorax.**
(XLSX)

## Author Contributions

**Conceptualization:** Yeong Hun Choe, Jeong-Hwan Hwang.

**Data curation:** Sang-Min Oh, Joo-Hee Hwang, Chang-Seop Lee.

**Formal analysis:** Sang-Min Oh, Joo-Hee Hwang, Chang-Seop Lee, Yeong Hun Choe, Jeong-Hwan Hwang.

**Investigation:** Sang-Min Oh, Joo-Hee Hwang, Chang-Seop Lee.

**Methodology:** Yeong Hun Choe, Jeong-Hwan Hwang.

**Supervision:** Yeong Hun Choe, Jeong-Hwan Hwang.

**Validation:** Yeong Hun Choe, Jeong-Hwan Hwang.

**Writing – original draft:** Yunhong Song, Jeongmin Lee, Wonchang Hahn, Yujeong Jang, Seungwon Na.

**Writing – review & editing:** Yunhong Song, Jeongmin Lee, Wonchang Hahn, Yujeong Jang, Seungwon Na.

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
