## [Decision Letter · Decision Letter 0]

11 Sep 2023

Dear Dr Hwang,

Thank you very much for submitting your manuscript "Risk factors and clinical features for pulmonary paragonimiasis-associated pneumothorax" for consideration at PLOS Neglected Tropical Diseases. As with all papers reviewed by the journal, your manuscript was reviewed by members of the editorial board and by several independent reviewers. In light of the reviews (below this email), we would like to invite the resubmission of a significantly-revised version that takes into account the reviewers' comments. 

We cannot make any decision about publication until we have seen the revised manuscript and your response to the reviewers' comments. Your revised manuscript is also likely to be sent to reviewers for further evaluation.

Sincerely,

Richard Stewart Bradbury, PhD

Academic Editor

Uriel Koziol

Section Editor

Reviewer's Responses to Questions

**Key Review Criteria Required for Acceptance?**

**Methods**

-Are the objectives of the study clearly articulated with a clear testable hypothesis stated?

-Is the study design appropriate to address the stated objectives?

-Is the population clearly described and appropriate for the hypothesis being tested?

-Is the sample size sufficient to ensure adequate power to address the hypothesis being tested?

-Were correct statistical analysis used to support conclusions?

-Are there concerns about ethical or regulatory requirements being met?

Reviewer #1: 1. The authors declare to evaluate frequency of paragonimiasis associated pneumothorax and assess risk factors.

2. The study design is inappropriate to test this hypothesis, as it is stated by the authors.

The authors should elaborate more on study methods: The study was stopped in 2017. was there a reason for that or it was arbitrary? further, the authors should expand on the definition of pulmonary paragonimiasis criteria and therefore inclusion criteria in their study.

Reviewer #2: The objectives are clearly articulated and the study design is appropriate. Statistical analysis was correctly used. No ethical or regulatory concerns were noted.

Reviewer #3: (No Response)

**Results**

-Does the analysis presented match the analysis plan?

-Are the results clearly and completely presented?

-Are the figures (Tables, Images) of sufficient quality for clarity?

Reviewer #1: 1. How was copd and asthma defined? were on treatment? only self reported? what was the cause, if available, or previous pneumothoraces?

2. Univariate analysis should be expanded on table 4 along with collineraity data. a separate table 5 with multivariable aanalysis of all variables even not significant ones should be created.

3. Collins and rea methodology for quantity of pleural effusion does not add anything in the manuscript and should be removed.

Reviewer #2: Results were clearly presented and supported by appropriate tables. May I suggest to consider renaming "incidence" as "positivity rate". Incidence is more appropriately used when considering a population under follow-up. Please elaborate on the possible reasons for relapse.

Reviewer #3: (No Response)

**Conclusions**

-Are the conclusions supported by the data presented?

-Are the limitations of analysis clearly described?

-Do the authors discuss how these data can be helpful to advance our understanding of the topic under study?

-Is public health relevance addressed?

Reviewer #1: The discussion section should be enriched. although these data are of significance and population stuydied is significant, the authors should expand on the cause of pneumothorax in their cases. Is it due to asthma, paragonimiasis, or both?

What possible implications have their findings in public health? future implications should be included in the discussion.

Reviewer #2: The conclusions are supported by the data presented. Limitations were properly cited. The manuscript discussed that the data are helpful to advance the understanding of the pneumothorax in pulmonary paragonimiasis.

Reviewer #3: (No Response)

**Editorial and Data Presentation Modifications?**

Reviewer #1: a better grammar is also needed. major revision required.

Reviewer #2: (No Response)

Reviewer #3: (No Response)

**Summary and General Comments**

Reviewer #1: This is a retrospective study with significant amount of paragonimiasis cases. Although results are interesting and the literature is well covered, i believe significant improvements should be made at methods section, results section and at discussion section., as point by point described above.

Reviewer #2: The manuscript is generally well written. The study assessed the clinical characteristics and risk factors of patients with pneumothorax associated with pulmonary paragonimiasis. The study highlights chest pain and a history of asthma as clinically significant risk factors. Please include pulmonary tuberculosis among the pulmonary diseases that mimic pulmonary paragonimiasis. A recommendation could be a larger cohort of pulmonary paragonimiasis with or without complications of pneumothorax possible with other centers in country or even in the region.

Reviewer #3: In this manuscript, the authors have described the risk factors and clinical features for pulmonary paragonimiasis-associated pneumothorax by a retrospective analysis of patients from their center. The data is important as they have touched upon a specific clinical feature of paragonimiasis, although the number of patients is low. However, since it is difficult to have more patients with this complication of pulmonary paragonimiasis, the number of patients is reasonable. 

The presentation of the data can be improved. The following are my suggestions.

Line 63-64: Modify as "... the cumulative incidence of pneumothorax was 13.6%..."

Line 69-70: The meaning is not clear. Please rephrase.

Line 126-127: The meaning is not clear. What do the authors mean by "All patients (aged ≥19 years) were diagnosed as having paragonimiasis between May 2011 and October 2017"? Please rephrase.

Line 129-130: Please provide details of the ELISA used? Please mention the source of the kit. 

How many patients were found positive by each of the modalities - ELISA, detection of eggs, detection of flukes?

Line 174-175: Please delete the statement about average height of the pneumothorax group greater than that of the non-pneumothorax group as it does not convey any useful information.

Line 193: Please use "eosinophil" instead of "eosinophile". Also change at other places in the manuscript.

Line 221-222: Do the authors want to say that "asthma (odds ratio [OR] 13.11 95% confidence interval [CI], 1.79-96.18) and chest pain (OR 7.41 95% CI, 2.22-24.78) were independent risk factors for pulmonary paragonimiasis-associated pneumothorax" or that asthma in patients with paragonimiasis has higher odds for developing pneumothorax. Similary, a patient of paragonimiasis with chest pain has higher odds of having developed pneumothorax?

Similarly, in the multivariate analysis, the other conditions were also found to have higher odds, i.e., previous pneumothorax and intrapulmonary lesion. These can also be mentioned in the text.

Line 238-239: Please rephrase.

Line 253: Please replace "larvae" with metacercariae or worm, as appropriate. Also change at other places in the manuscript.

PLOS authors have the option to publish the peer review history of their article (what does this mean?). If published, this will include your full peer review and any attached files.

Reviewer #1: Yes: Ilias Papanikolaou

Reviewer #2: No

Reviewer #3: No

Figure Files:

Data Requirements:

Please note that, as a condition of publication, PLOS' data policy requires that you make available all data used to draw the conclusions outlined in your manuscript. Data must be deposited in an appropriate repository, included within the body of the manuscript, or uploaded as supporting information. This includes all numerical values that were used to generate graphs, histograms etc.. For an example see here: http://www.plosbiology.org/article/info:doi%2F10.1371%2Fjournal.pbio.1001908#s5.
---

## [Editor Report · Decision Letter 1]

23 Nov 2023

Dear Dr Hwang,

Thank you very much for submitting your manuscript "Risk factors and clinical features for pulmonary paragonimiasis-associated pneumothorax" for consideration at PLOS Neglected Tropical Diseases. As with all papers reviewed by the journal, your manuscript was reviewed by members of the editorial board and by several independent reviewers. The reviewers appreciated the attention to an important topic. Based on the reviews, we are likely to accept this manuscript for publication, providing that you modify the manuscript according to the review recommendations. 

This revised version of the manuscript is much improved. We particularly commend the authors on extending the retrospective study dates to 2021. Some minor elements still require minor amendment, specifically;

1) Plese use "eosinophil" instead of "eosinophile" throughout

2) Lines 291, 299, 305. At the stage of the life cycle where Paragonimus are migrating to the lungs, they are correctly called "larvae" (stage between excystation of the metacercariae in the duodenum and maturation to adulthood once established in the lung). Please replace "worm" with "larvae" at these three points (we apologise for prior incorrect instruction regarding this).

3) Line 322 please change to "the giant intestinal roundworm"

4) Please italicise genus and species names in your reference list

Sincerely,

Richard Stewart Bradbury, PhD

Academic Editor

Uriel Koziol

Section Editor

This revised version of the manuscript is much improved. We particularly commend the authors on extending the retrospective study dates to 2021. Some minor elements still require minor amendment, specifically;

1) Plese use "eosinophil" instead of "eosinophile" throughout

2) Lines 291, 299, 305. At the stage of the life cycle where Paragonimus are migrating to the lungs, they are correctly called "larvae" (stage between excystation of the metacercariae in the duodenum and maturation to adulthood once established in the lung). Please replace "worm" with "larvae" at these three points (we apologise for prior incorrect instruction regarding this).

3) Line 322 please change to "the giant intestinal roundworm"

4) Please italicise genus and species names in your reference list

Figure Files:

Data Requirements:

Please note that, as a condition of publication, PLOS' data policy requires that you make available all data used to draw the conclusions outlined in your manuscript. Data must be deposited in an appropriate repository, included within the body of the manuscript, or uploaded as supporting information. This includes all numerical values that were used to generate graphs, histograms etc.. For an example see here: http://www.plosbiology.org/article/info:doi%2F10.1371%2Fjournal.pbio.1001908#s5.

Reproducibility:

References

---

## [Editor Report · Decision Letter 2]

28 Nov 2023

Dear Dr Hwang,

We are pleased to inform you that your manuscript 'Risk factors and clinical features for pulmonary paragonimiasis-associated pneumothorax' has been provisionally accepted for publication in PLOS Neglected Tropical Diseases.

Best regards,

Uriel Koziol

Section Editor

Uriel Koziol

Section Editor

---

## [Editor Report · Acceptance letter]

8 Dec 2023

Dear Dr Hwang,

We are delighted to inform you that your manuscript, "Risk factors and clinical features for pulmonary paragonimiasis-associated pneumothorax," has been formally accepted for publication in PLOS Neglected Tropical Diseases.

Best regards,

Shaden Kamhawi

co-Editor-in-Chief

Paul Brindley

co-Editor-in-Chief
